# Gene and allele-specific expression during electric organ ontogeny in African weakly electric fish (Campylomormyrus)
Feng Cheng [1], Alice B. Dennis[1,2], Linh Nguyen[1], Otto Baumann[3], Frank Kirschbaum[1,4], Marisol Domínguez[1] & Ralph Tiedemann [1] ✉

The African weakly electric fish use their muscle-derived electric organ to produce electric organ discharge (EOD) for electrocommunication and electrolocation. The EOD among species of the genus *Campylomormyrus* and cross-species hybrids is species-specific and varies during ontogeny. We compared the gene expression patterns and allele-specific expression between juvenile and adult in *C. compressirostris* (EOD duration 0.4 ms in both juveniles and adults), *C. rhynchophorus* (EOD duration 5 ms in juveniles and 40 ms in adults) and their hybrids (EOD duration 0.4 ms in juveniles and 4 ms in adults). By clustering the gene expression and EOD duration, we identified several candidate genes that might be involved in EOD development, i.e., *KCNJ2*, *CPNE7* and *CADPS*. In the hybrids, the alleles from parental species show nearly equal expression at both juvenile and adult life stages. *KCNJ2* exhibits allelic dominance of the *C. rhynchophorus* allele, with further increasing expression during ontogeny. This suggests that the EOD development in hybrids could be related to the increasing allelic expression of the *C. rhynchophorus* allele. Our study sheds light on the evolution and ontogeny of the EOD in electric fishes and on the potential of hybridization to understand the molecular underpinning of complex phenotypic traits.

Context-dependent differential gene expression throughout ontogeny contributes to the formation of complex phenotypes; hence, divergence in complex traits among closely related species may be understood by investigating differential gene expression during ontogeny. In addition, hybrids among such target species—should they be viable—can play an important role in understanding the emergence of complex phenotypes[1]. Experimental hybrids offer opportunities to study phenotypes that differ from traits in the purebred species[2,3]. One possible way to find the underlying genes of adaptive traits diverging among species is to detect allele-specific expression in hybrids, i.e., an imbalance between the expression levels of the alleles from the two respective parental species. Allele-specific expression has been identified in hybrids and polyploids of many animals and plants[4–8]. In the fish hybrid of *Megalobrama amblycephala* and *Culter alburnus*, the genome-wide transcriptional analysis revealed that the asymmetric expression of alleles could counteract the deleterious effects of the sub-genomes (half genome from each parental species), and hence improve the adaptability of novel hybrids[9].

The African weakly electric fishes (Mormyridae) possess a "magic trait" to produce and perceive electric signals[10]. Mormyrids use their electric organ to generate an electric organ discharge (EOD) for object sensing and electrocommunication in a social context, including mate choice. The EOD is typically species-specific and varies in shape and waveform, which is considered as a pre-zygotic isolation mechanism in mormyrids[10,11]. The EOD in mormyrids is a compound action potential from the simultaneously firing electrocytes. The depolarization and repolarization phases of an action potential are mediated by multiple, interacting, inward and outward ion currents[12–14]. In *Campylomormyrus*, the diverged EOD waveform might relate to the strength of potassium currents. The inwardly rectifying potassium channel *Kir2.1*, which is encoded by the gene *KCNJ2*, possibly regulates the action potential duration and exhibits significantly higher expression in species with elongated EOD[15]. In addition, the voltage-gated potassium channel gene *KCNA7A*, which mediates outward potassium currents, also potentially contributes to the EOD duration in weakly electric fish, as evidenced in the mormyrid genera *Gymnarchus* and *Brienomyrus*[16].

[1]Unit of Evolutionary Biology/Systematic Zoology, Institute of Biochemistry and Biology, University of Potsdam, Potsdam, Germany. [2]Laboratory of Adaptive Evolution and Genomics, URBE, Institute of Life, Earth & Environment, University of Namur, Namur, Belgium. [3]Department of Animal Physiology, Institute of Biochemistry and Biology, University of Potsdam, Potsdam, Germany. [4]Department of Crop and Animal Science, Faculty of Life Sciences, Humboldt University, Berlin, Germany. ✉e-mail: tiedeman@uni-potsdam.de

The genetic background of EOD diversification has been studied in several genera, while their development is still unknown. The EOD does not only vary among different species, but also in different life stages in *Campylomormyrus*[17–19]. Early in ontogeny, they possess a larval electric organ in the deep lateral muscle, which is later reduced, while the persistent adult electric organ is located in the caudal peduncle. Developmental change in electrocytes includes cell growth, formation of papillae (in *C. rhynchophorus*, *C. numenius*, and hybrids *C. compressirostris* x *C. rhynchophorus*), and other structural aspects[17,19]. Previous investigations on the ontogeny of the EOD in the genus *Campylomormyrus* species showed that *C. compressirostris*, *C. tamandua*, *C. tshokwe*, and *C. rhynchophorus* share short-duration EODs in their early juvenile stage. In *C. compressirostris*, EOD stays consistent during development, while the other studied species undergo multiple alterations until they produce an adult EOD[18]. Developmental EOD variation was also found in other mormyrid species, e.g., *Mormyrops* and *Paramormyrops*[17].

Successful breeding experiments have been performed in *Campylomormyrus* species as well as interspecific hybrids could be produced[18], although such hybrids are rarely found in the natural habitat. Taking advantage of the experimental hybridization, we are able to observe the EOD variation in purebred species and hybrids during ontogeny. *Campylomormyrus* hybrids also start with similar short-duration juvenile EOD, and reach an adult EOD after several changes in juvenile stages. However, regarding the EOD the apomorphic feature (medium and long duration) is always dominant over the plesiomorphic (short duration) one. The adult EOD of hybrid *C. compressirostris* x *C. rhynchophorus* is similar to the EOD of 6 cm-sized juveniles of *C. rhynchophorus*[18].

If the species-specific EOD contributes to the isolation mechanism for *Campylomormyrus*, it might be fixed among species. This study is trying to understand the molecular underpinning of EOD differences from a developmental perspective. Our objective is to address the role of hybridization-derived genetic architecture on the expression of complex phenotypes (here, EOD) by identifying genes contributing to EOD development during the ontogeny of parental species, as well as assessing allele-specific expression during hybrid development (Fig. 1). We sampled two life stages (juveniles around 7 cm body length and adults) from two F0 species and their F1 hybrid (Fig. 1a): *Campylomormyrus compressirostris* (*com*, 0.4 ms ["short"] biphasic juvenile and adult EOD, i.e., no change in EOD during ontogeny), *C. rhynchophorus* (*rhy*, 5 ms ["medium"] biphasic juvenile EOD, 40 ms ["long"] triphasic adult EOD,), and their hybrid *C. compressirostris* ♂ x *C. rhynchophorus* ♀ (*com* x *rhy*, 0.4 ms ["short"] biphasic juvenile EOD; 4 ms ["medium"] biphasic adult EOD).

## Results

The overall gene expression patterns from both juvenile and adult RNA-seq were analyzed by principal component analysis (PCA, Fig. 2a). This PCA plot supports a distribution pattern based on EOD duration in PC1, which explains 32% of the variation. The expression pattern from both adult and juvenile *com* samples was clustered together, corresponding with the EOD consistency during ontogeny in this species. In contrast, the adult EODs in the *rhy* and F1 hybrid exhibit shape variation and duration elongation compared with juvenile EODs, and there is a separation between juvenile and adult RNA-seq data in the PCA plot for these samples.

### Genes related to electric organ development during ontogeny

Since the PCA plot (Fig. 2a) exhibited clear association between EOD duration and overall gene expression during ontogeny, we performed a likelihood ratio test (LRT) implemented in DESeq2 to identify genes with significant associations between their expression and EOD duration[20] (Fig. 1c). The LRT analysis ascertained 7094 genes that showed significant differential gene expression over the different EOD durations using a threshold of adjusted *P* value < 0.05. Using the degPatterns function, we

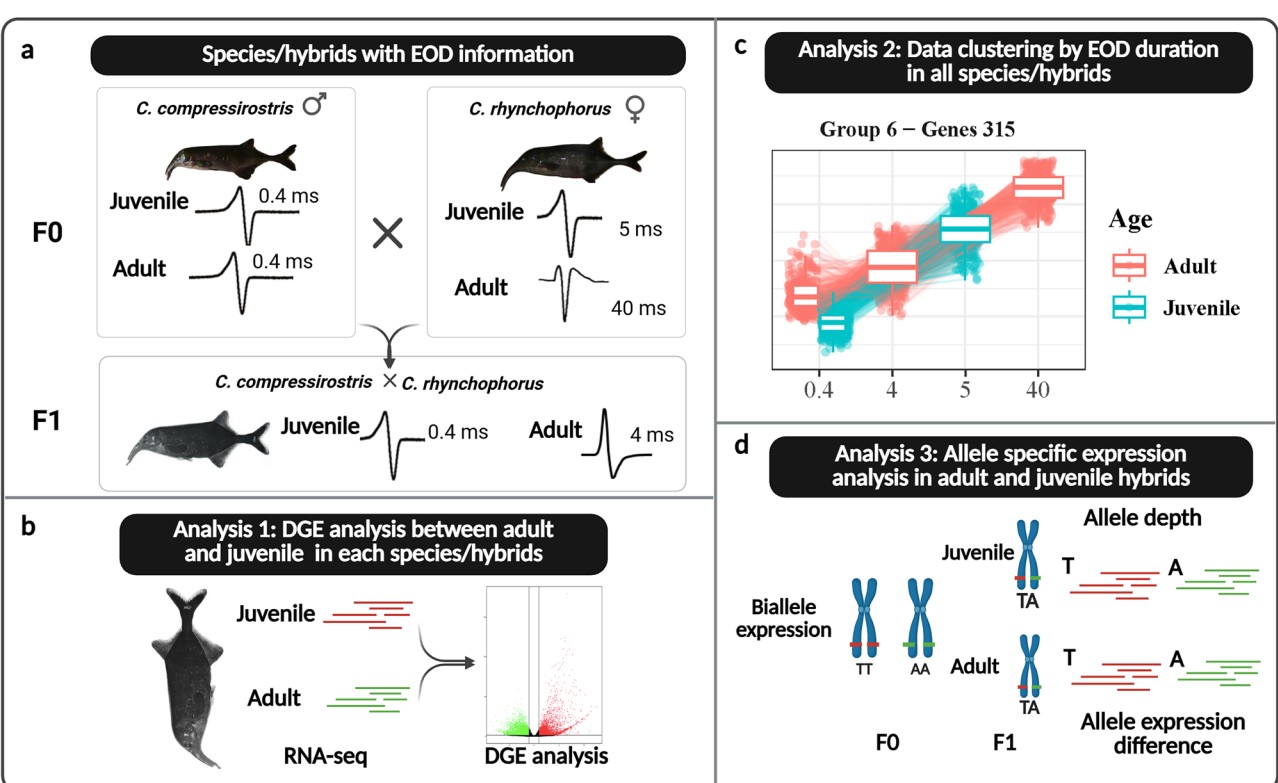

**Fig. 1 | Electric organ discharge (EOD) of the studied *Campylomormyrus* species and hybrids, and working flow in this study. a** Species/hybrids samples used in the study have EODs with different durations and shapes. **b** Differential gene expression (DGE) analysis between juveniles and adults for each species/hybrid. **c** RNA-seq data clustering by EOD duration in all species/hybrids. **d** Allele-specific expression analysis in adult and juvenile hybrids. Only homozygous bialleles in F0 were considered, and the allelic depth was counted to assess allele-specific expression in two life stages of hybrids.

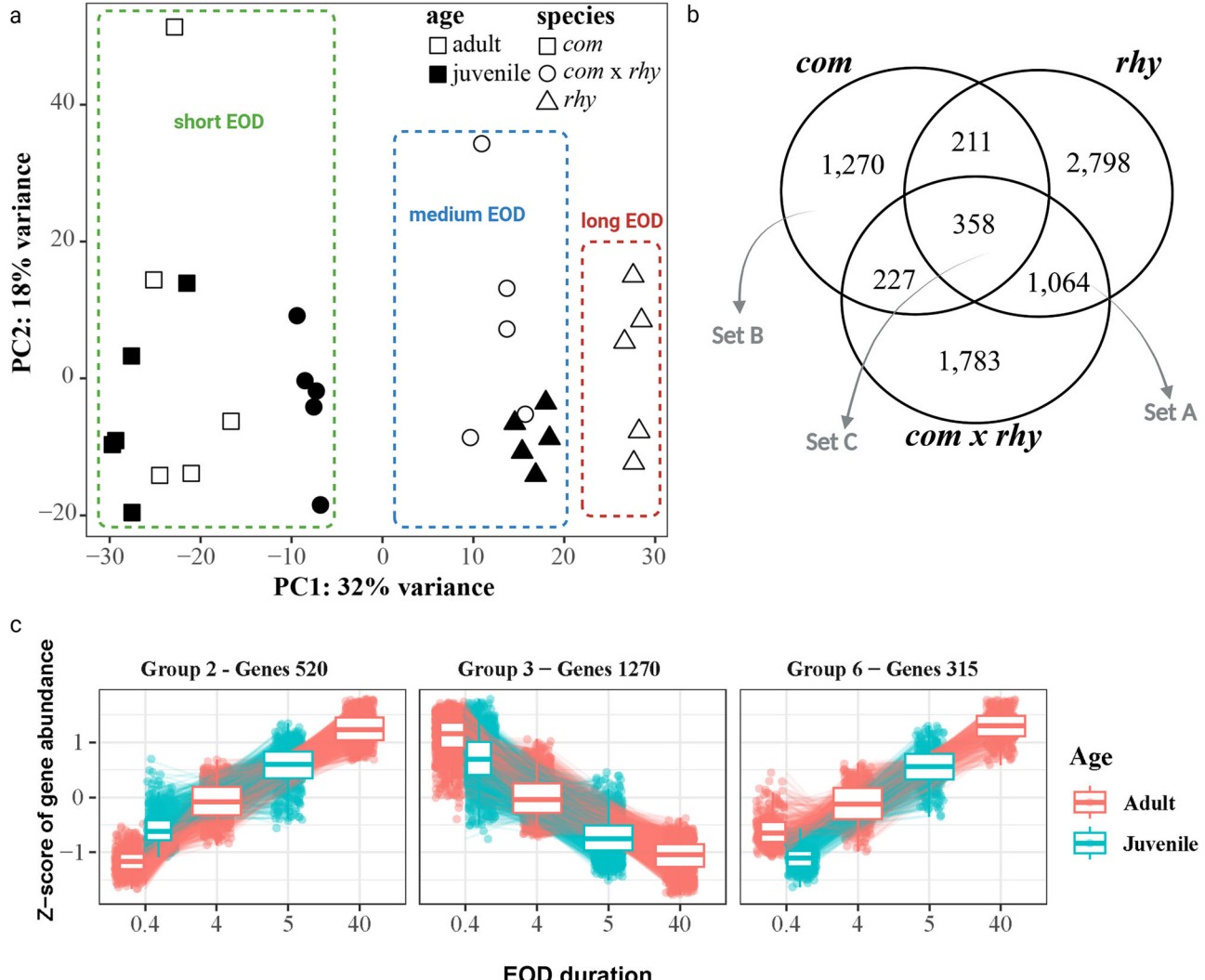

**Fig. 2 | Gene expression analysis of the comparison between adult and juvenile species/hybrids, and data clustering by EOD duration. a** Principal Component Analysis (PCA) of gene expression levels in all juvenile and adult species/hybrids. **b** Venn Diagram graph for differentially expressed genes (DEGs; |log2(fold change) | > 1; $P < 0.05$) between juvenile and adult shared among the three species/hybrids. **c** RNA-seq data clustered by EOD duration.

received 14 groups of different gene expression patterns in adults and juveniles, relative to EOD duration (Supplementary Fig. 1). We focused on the groups which showed an increasing or decreasing pattern of gene expression, relative to EOD duration (Fig. 2c). Group 2 and 6 (520 and 315 genes, respectively) exhibited increasing patterns while group 3 (1270 genes) showed a decreasing pattern that relative to EOD duration.

We also performed gene expression pairwise comparisons between adult and juvenile electric organs for each species and the hybrids separately (Fig. 1b). We specifically looked at the differentially expressed genes (DEGs) that overlapped among different species/hybrid (Fig. 2b). In *rhy* and the hybrids *com* x *rhy*, the EOD exhibits shape change and duration elongation from juvenile to adult, while the EOD does not change during ontogeny in *com*. Therefore, we focused on three sets of DEGs: set A contains the shared DEGs only in *rhy* and hybrid, representing the DEGs potentially associated to EOD change during ontogeny; set B contains the DEGs unique to *com*, representing the DEGs corresponding to an EOD consistency during ontogeny; set C contains the DEGs shared in all species and the hybrids, representing the DEGs corresponding to general electric organ development. In total, we identified 1064 DEGs in set A, 1270 in set B, and 358 in set C (Fig. 2b).

Among the EOD development candidate genes (2105 genes from group 2, 3, and 6), we searched for genes present in the following gene lists: (i) DEGs of set A, (ii) DEGs of set B, (iii) DEGs of set C, and two lists from Cheng et al. 2024, i.e., (iv) upregulated genes in adult EOs compared to skeletal muscles, and (v) EOD duration candidate gene set in adult EOs. Of the 2105 genes, 119 genes were also listed in set A and B, and those genes were significantly enriched in "structural molecule activity", "actin binding", and "calcium ion binding" Gene Ontology terms (Fisher's exact test $P$ values are 0.002, 0.021, and 0.045, respectively; Supplementary Tables 1 and 2). Seven genes, *RYBP*, *MIPOL1*, *KCNJ2*, *CPNE7*, *CLDN11*, *CPT2*, and *CADPSA*, were present in three of the gene sets (Table 1) and are hence strong candidates to be involved in the EOD development during ontogeny.

**Allele-specific expression during ontogeny**

For genes where the purebred parental species exhibited fixed (homozygous) differences, allele-specific expression analysis was performed (Fig. 1d). In total, we identified fixed SNPs in 343 genes (Fig. 3a) which were significantly enriched in "cytoplasm" and "cytoskeleton" Gene Ontology terms ($P$ value < 0.001, Supplementary Fig. 2). Among them, 217 genes in juveniles and 204 in adults had an equal expression of both parental alleles (i.e., expression proportion of *com* allele around 0.45–0.55), representing 63.3% and 59.5% of the genes, respectively (Fig. 3a). Among the genes that showed unequal expression (expression proportion outside 0.45–0.55),

**Table 1 | Candidate genes that might relative to EOD development during ontogeny**

| Group | ID | Datasets | | | | | | | Blast | Category |
|---|---|---|---|---|---|---|---|---|---|---|
| | | Gene | Upregulated in adult EOs (from Cheng et al.[15]) | EOD duration candidates in adult EOs (from Cheng et al.[15]) | Set A | Set B | Set C | Group 2, 3, or 6 | | |
| 3 | maker-ptg000119l-snap-gene-11.24-mRNA-1 | CADPS | ✓ | ✓ | ✓ | | | ✓ | Calcium-dependent secretion activator 1 | Signaling |
| 2 | maker-ptg001737l-snap-gene-0.24-mRNA-1 | CLDN11 | ✓ | ✓ | | ✓ | | ✓ | Claudin-11 | Other |
| 3 | maker-ptg000950l-augustus-gene-2.9-mRNA-1 | CPNE7 | ✓ | ✓ | ✓ | | | ✓ | Copine-7 | Transmembrane ion transport |
| 3 | maker-ptg000744l-snap-gene-3.34-mRNA-1 | CPT2 | ✓ | ✓ | | ✓ | | ✓ | Carnitine O-palmitoyltransferase 2, mitochondrial | Other |
| 2 | maker-ptg000265l-est_gff_est2genome-gene-6.33-mRNA-1 | KCNJ2 | ✓ | ✓ | ✓ | | | ✓ | Inward rectifier potassium channel 2 | Transmembrane ion transport |
| 3 | maker-ptg000051l-snap-gene-29.0-mRNA-1 | MIPOL1 | ✓ | ✓ | | ✓ | | ✓ | Mirror-image polydactyly 1 | Other |
| 3 | maker-ptg001058l-augustus-gene-1.50-mRNA-1 | RYBP | ✓ | ✓ | | ✓ | | ✓ | RING1 and YY1-binding protein B | Other |

there were almost equal numbers exhibiting a preferential expression of the *com* allele and the *rhy* allele.

In all, 39 genes exhibited an allelic expression proportion change over 0.1 (Supplementary Table 3). Three of those genes were also in the list of potential candidate genes of group 2, 3, and 6 (derived from the cluster analysis): *KCNJ2*, *HIP1*, and *PPP1R3A* (Supplementary Tables 2 and 3).

We specifically looked at the allelic expression in each individual at gene *KCNJ2*, since the expression of this gene has been found to be upregulated in *Campylomormyrus* species with elongated EOD duration, and it is a strong candidate for EO development (Table 1). In this study, the *com* allele of *KCNJ2* gene showed a higher expression proportion (0.30) in the juvenile hybrid than in the adult (0.20; *t* test of *com* proportion $P = 0.026$; Fig. 3b). The *rhy* allele in *KCNJ2* gene is more dominant in the adult electric organ, indicating a considerable allelic expression shift during hybrid ontogeny. Apart from gene *KCNJ2*, another gene (*SCN4AA*) also exhibited high allelic expression change during ontogeny. Two alleles were nearly equally expressed in the juveniles (*com* proportion = 0.55), while the adults had significantly higher expression of *com* alleles (0.67; *t* test of *com* proportion $P = 0.002$; Fig. 3b).

## Discussion

### Electric organ discharge development during ontogeny in *Campylomormyrus*

Knowledge of phenotypic development can greatly contribute to our understanding of the mechanisms underlying phenotypic divergence in natural populations. The EOD divergence is considered a strong pre-zygotic isolation mechanism among *Campylomormyrus* species. Understanding the ontogeny of this key speciation trait might help us to unravel the mechanism behind their ecological diversification. The depolarization and repolarization during an EOD are associated with sodium and potassium currents across the plasma membrane of the electrocytes. In electric fish, the initial depolarization phase is mainly driven by the inward sodium current through voltage-gated sodium channels, and repolarization, ultimately returning the membrane to the resting potential, is principally affected by the outward current through voltage-gated potassium channels[12].

The weakly electric mormyrids usually possess a larval electric organ at an early age and an adult electric organ that persists in adults. Among *Campylormormyrus* adults, EODs are generally different from the larval and juvenile EODs during ontogeny. For example, *C. rhynchophorus* (*rhy*) possesses a relatively short biphasic juvenile EOD (0.5 ms), and gradually reaches an adult long triphasic EOD (40 ms) during development. Deviating from this, the EOD in *C. compressirostris* (*com*) shows consistency during ontogeny, remaining very short (0.4 ms) and biphasic. In hybrids *com* x *rhy*, the early juvenile EOD is quite similar in shape and duration to the *com* EOD. In adulthood, however, the hybrid EOD was still biphasic but elongated (5 ms), resembling the EOD of a *rhy* juvenile of ~6-7 cm body length. This suggests that the EOD development in these hybrids is closer to the putatively apomorphic elongated *rhy* EOD.

The gene *KCNJ2* was already identified as a powerful candidate gene in regulating the EOD duration, with higher expression in the adult EOs in long EOD species[15]. Here, we also found that this gene might affect the EOD elongation during development in *rhy* and the hybrid *com* x *rhy* (Fig. 2c and Table 1) since it exhibited higher expression in the adult EOs compared to juvenile EOs (in line with shorter EOD in juveniles compared to adults). In *com*, its expression revealed no significant difference between juveniles and adults, in line with the unchanged EOD throughout the entire ontogeny in this species. The inward potassium channel encoded by *KCNJ2* gene can pass positive charged current more easily in the inward than the outward direction[21,22]. This channel is considered as important in regulating neuronal activity by stabilizing the resting membrane potential. Moreover, it may contribute to the shaping of the initial depolarization and the final repolarization step during the action potential, as found in human cardiomyocytes[21].

Apart from the *KCNJ2* gene, *CADPS* and *CPNE7* might also contribute to EOD elongation during development. *CADPS* encodes for a calcium-

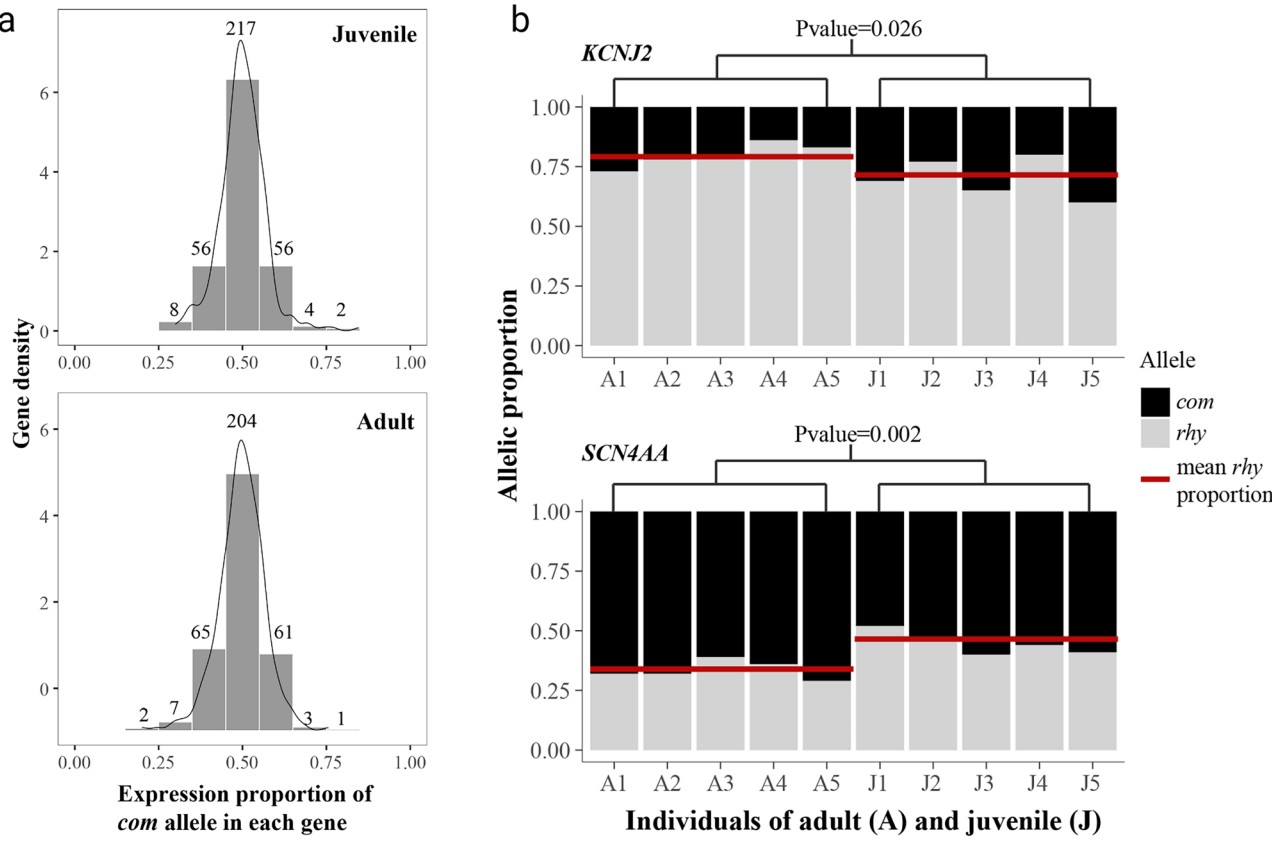

**Fig. 3 | Proportion of *com* allele expression, and *com* allele expression change between juvenile and adult hybrids. a** Gene density (*y* axis) from juvenile and adult stages in the hybrid. The *x* axis shows the expression proportion of the allele stemming from one parental species (*com*). Numbers above the bars represent the number of genes in the respective proportion ranges. **b** Allelic proportions of the *KCNJ2* and *SCN4AA* genes in each individual of juvenile and adult hybrids.

dependent activator protein that functions in the release of catecholamine in neuroendocrine cells[23]. *CPNE7* encodes a member of the copine family, which is composed of calcium-dependent membrane-binding proteins, and may be involved in membrane trafficking[24]. It regulates axonal/dendritic extension in other cells. However, the function of these genes in the development of the electric organ still needs to be tested.

The expression of genes *CLDN11*, *CPT2*, *MIPOL1*, and *RYBP* stayed consistent in *com* during development, while their expression exhibited either an increasing or decreasing pattern when EOD got elongated (Fig. 2c). *CLDN11* encodes for a member of the claudin family, which are integral membrane proteins and components of tight junction strands[25]. The disruption of *CLDN11* in mice may impair saltatory conduction and alter the speed of action potential in neurons[25]. In the genomes of teleost fishes, a large number of genes encoding for claudins exist, resulting from whole genome duplication and multiple tandem gene duplication events[26]. Claudins are detected in different tissues of teleost fishes, i.e., liver, brain, and heart. The expression of *CLDN11* is spotted in myelinated portions of the optic nerve of tilapia[27]. Many claudin members are reported to play vital roles in the embryonic development of vertebrates[25].

These results support the notion that electrocytes in different species/ hybrids and/or at a different developmental stage are equipped with a unique set of transcribed genes, which might be critical for the duration (and possibly also the shape) of the specific EOD. Apart from the mentioned genes, other genes might also contribute to the EOD development, i.e., potassium channel genes (*KCNA7A_1*, *KCNQ5A*) and gelsolin (*GSN*; Supplementary Table 2). The exact function of these potassium channels in this scenario must remain open, since detailed information of the electrophysiological properties of the various channel types in these species, e.g., the effects of a different subunit composition for a distinct channel type, are missing. Further studies about the histology and physiology of electric organ

development would be essential to understand the mechanisms of EOD development.

**Allele-specific expression reveals subgenome effect in hybrids**

Adaptive divergence often involves changes in multiple traits to improve reproductive fitness in a particular environment. Traits may be fixed within populations if they play key roles in adaptation and speciation. Experimental hybridization provides the opportunity to study new or intermediate traits, owing to the expression of two subgenomes from two species/populations[2,3]. As the key trait for speciation (here the EODs) is invariant within *Campylomormyrus* species, species may be fixed for different alleles at underlying loci.

Around 60% of the fixed bialleles showed almost balanced expression of the alleles of both parents, both in the juvenile and adult stages of hybrids. Among the genes that showed unbalanced expression, there was an almost equal number of genes with higher expression of the respective allele of either parental species (Fig. 3a). During the development of the hybrids, many genes with fixed differences among parental species exhibited obvious expression changes, relative to their parents (Supplementary Table 3). We noticed that genes *KCNJ2*, *HIP1*, and *PPP1R3A* exhibited expression proportion changes over 0.1 during development. These genes were also inferred as candidates for EOD development (in groups 2, 3, and 6; Fig. 2c). All three genes showed a higher expression proportion on *rhy* alleles in the adults, compared to juveniles.

*KCNJ2* showed a significant expression dominance of the *rhy* allele in the *com* x *rhy* hybrid at both juvenile and adult stages, and the dominance was increasing during ontogeny (Fig. 3b). In pairwise comparison of gene expression between the F1 hybrids and each parental species, *com* showed down-regulation compared to the hybrid in both life stages, while *rhy* showed up-regulation (Supplementary Table 3). This suggests that this gene might be under *cis*-regulation. The genes *HIP1* and *PPP1R3A* did not show

such a pattern. As a strong candidate for affecting EOD duration among species and EOD elongation during development, *KCNJ2* might affect the EOD alteration in duration (perhaps shape as well) throughout the development of the hybrids.

Two juvenile individuals (J2 and J4 in Fig. 3b) showed a *rhy* allele expression proportion for *KCNJ2* similar to the adult. These juvenile individuals already showed a slightly elongated EOD (around 0.45 ms, relative to 0.4 ms in the other juveniles). It is likely that the increased expression of the *rhy* allele occurred when the juvenile EODs started to elongate, and stayed at similar proportions until adulthood, further suggesting that this gene triggers EOD elongation during ontogeny.

The F1 hybrids *com* x *rhy* contain half genomes (subgenomes) from each parental species. Although the genes with fixed differences analyzed here do not represent the full subgenomes, our allele-specific expression analysis still gives us a hint that the subgenomes might generally have an equal impact on the F1 hybrids during ontogeny from the expression perspective. Nonetheless, the development of a particular phenotypic feature (here EOD elongation) is likely affected by the imbalanced allelic expression of the potentially *cis*-regulated gene *KCNJ2*. The physiological function of this gene is not well studied in the electric organ, but a knock-out/down experiment would be beneficial to understanding its impact.

## Conclusion

An important driver of divergent phenotypic evolution in closely related species is the expression divergence in underlying genes. For a particular trait, F1 hybrids may exhibit intermediate phenotypes, as they express alleles of two subgenomes from their parental species, or may preferentially resemble the trait of one parent (dominance). In our hybrids, the phenotype (i.e., the EOD) is more similar to one parent (*C. compressirostris*) in juveniles, but to the other parent (*C. rhynchophorus*) in adults.

Allele-specific expression analysis in hybrids proved a generally equal expression of alleles from both parental species, while the EOD phenotype was closer to one parental species (*C. rhynchophorus*). This could point to the involvement of a few major genes in EOD alteration. Indeed, *KCNJ2* showed significant allelic expression dominance of the *C. rhynchophorus* allele in both juvenile and adult life stages, and was increasingly dominant during ontogeny. This gene was previously assumed to be involved in the EOD duration regulation in *Campylomormyrus*[15]. In addition, its expression was also increased throughout the ontogeny of *C. rhynchophorus* and the hybrid, both of which underwent a prolongation in EOD duration during ontogeny. Therefore, we hypothesize *KCNJ2* to be a crucial gene in regulating the EOD duration during the ontogeny of *Campylomormyrus*. Moreover, it likely affects the EOD in the hybrid development.

A functional test of the *KCNJ2* gene in electric fish is lacking, but would be essential to prove its role in EOD divergence and development. In addition, a potential backcross experiment of F1 hybrid with *C. compressirostris* might be interesting to explore whether these backcrosses would exhibit different EODs in relation to allelic state and expression of *KCNJ2*.

It should be noted that our differential gene expression and allele-specific expression approach is tailored towards the identification of candidate genes with strong expression differences and/or fixed differences, relative to divergence in a complex phenotype (here, duration of the electric organ discharge). We argue that— given that the EOD profoundly differs among closely related species, but is almost invariant within species of *Campylomormyrus*—the assumption of fixed genetic differences among species, be it in protein-coding exons or gene-regulatory elements, appears reasonable. Indeed, we identify strong candidates for such effects here. Nonetheless, inheritance of the EOD may be polygenic, and our approach may overlook involved genes with less profound differential expression and/ or without fixed differences among species/EOD phenotypes.

## Methods
### Samples and RNA sequencing
We sampled the electric organ samples from two F0 species (*C. compressirostris*, *com*; *C. rhynchophorus*, *rhy*) and F1 hybrids (*C. compressirostris*

♂ x *C. rhynchophorus* ♀) at the juvenile (around 6 to 7 cm body length) life stage. The juvenile species/hybrids were artificially bred and raised at the University of Potsdam. We have complied with all relevant ethical regulations for animal use. We included five individuals from each species/hybrid cohort for mRNA sequencing. We anesthetized the fish specimens using a lethal dose of clove oil and sampled the electric organs within 99% ethanol above ice. The extracted electric organ samples were flash-frozen in liquid nitrogen and preserved in −80 °C.

We applied QIAGEN RNeasy Fibrous Tissue Kit on all juvenile electric organ samples for RNA extraction, and further inspected the quantity and quality using a NanoDrop 1000 spectrophotometer (ThermoFischer Scientific, Germany) and an Agilent Bioanalyzer 2100 (Agilent Technologies, USA), respectively. We performed mRNA enrichment via poly (A) capture from the isolated RNA (RIN > 9.0 in Bioanalyzer) using NEXTflex Poly (A) Beads, and then established the strand-specific transcriptomic libraries through NEXTflex Rapid Directional RNA-Seq Kit (Bioo Scientific, USA).

All constructed libraries were sent to a commercial company (Novogene) for sequencing of 150 bp paired-end reads using an Illumina HiSeq 4000 sequencing system. Raw reads were deposited in the National Center for Biotechnology Information (NCBI) Gene Expression Omnibus.

We also included the mRNA sequences of electric organs of adult *C. compressirostris*, *C. rhynchophorus* and their hybrid *C. compressirostris* ♂ x *C. rhynchophorus* ♀ from Cheng et al.[15] (accession number: GSE240783). In order to remove the adapter sequences and low-quality reads, all sequences were trimmed using a 4 bp sliding window with a mean quality threshold of 25, and a minimum read length of 36 bp by Trimmomatic v0.39[28]. Read quality was measured by FastQC v0.11.9[29].

### Differential gene expression analysis
We mapped the quality-filtered RNA reads from each sample to the *C. compressirostris* genome by RSEM[30]. The estimated gene-level quantification counts were imported to R/Bioconductor using the tximport package. We removed genes with low count (≤ 10) and in less than two replicates to get cleaned reads. A Principal Component Analysis (PCA) was performed on the cleaned and log-transformed counting matrices (using covariance).

For the purpose of identifying genes that were differentially expressed between adult and juvenile samples, we applied normalized counting matrices to DESeq2 between adults and juveniles of the same species/hybrid cohort[31]. Differentially expressed genes (DEGs) were identified using |log2 Folder change (log2FC)| >1 and a *P* value < 0.05. A Venn Diagram was compiled to visualize shared DEGs across species/hybrid cohorts[32]. We focused on three DEGs sets: shared DEGs among fish with EOD elongation throughout ontogeny (*rhy*, hybrids), set A; DEGs among fish with EOD constancy throughout ontogeny (*com*), set B; and shared DEGs among all analyzed electric fish (*com*, *rhy*, hybrids), set C. These three sets of DEGs were blasted against the *nr* database from NCBI using blastx with an e-value cutoff $1e^{-10}$.

### Identification of candidate genes for EOD duration
Since the PCA showed a strong association between EOD duration and gene expression pattern in PC1, we performed a likelihood ratio test (LRT) implemented in DESeq2 to identify genes with an expression pattern associated to EOD duration[20]. We applied a "full model" with "~EOD duration" compared to a "reduced model" (here null). Genes showing significant association with EOD duration were extracted using an adjusted *P* value of 0.05. We used the degPatterns function to cluster particular gene expression patterns of those significant genes across EOD duration and ontogenetic stages[31].

Gene expression clusters exhibiting a consistent increasing or decreasing pattern with increasing EOD duration were considered to identify candidate genes (i.e., genes up-/down-regulated, relative to EOD duration). Gene Ontology enrichment analysis was performed on those genes using the genome of zebrafish as background. In addition, we evaluated whether those genes were in the following gene lists: (i) DEGs of set A, (ii) DEGs of set B, (iii) DEGs of set C, and two lists from Cheng et al.[15], i.e.,

(iv) upregulated genes in adult EOs compared to skeletal muscles and (v) EOD duration candidate gene set in adult EOs. We blasted genes represented in at least one of these datasets against the *nr* database from NCBI (using blastx with an *e*-value cutoff $1e^{-10}$) to infer gene identity.

## Allelic-specific expression analysis between adult and juvenile hybrids

In this study, we focused on genes containing SNPs with fixed bialleles (homozygous) in the parental F0 species. We mapped the cleaned mRNA reads from each sample to *C. compressirostris* genome by STAR v2.7.7 + WASP[33], and sorted the generated bam files from STAR according to the coordinates using SAMtools v1.15[34]. All the sorted bam files were imported to BCFtools v1.9 for variant calling using "bcftools mpileup –f <*C. compressirostris* CDS genome > <list of bam files> –Ou | bcftools call –mv –Ob –o <output bcf file > "[34]. We only preserved the fixed parental biallelic SNPs for both juvenile and adult hybrids, and calculated the allelic depth by "bcftools view --exclude-uncalled -m2 -M2 < bcf file > > <allelic depth > "[34]. We filtered out the SNPs that with QUAL score lower than 70 as well as with an allelic depth lower than $10^{15}$.

Eventually, we acquired high-quality biallelic SNPs for both adult and juvenile hybrids. Expression proportion of the alleles from *C. compressirostris* (com) were calculated for each sample, and further merged at gene level by calculating the mean proportion if there were more than one SNP located in one gene. We calculated the average proportion of each sample in both juvenile and adult stages in the hybrids, as well as the proportion alteration during ontogeny. For genes with fixed SNPs, we performed the pairwise comparisons between each parental species and F1 hybrids, both for juvenile and adult stages. The resulting log2FC values are provided in Supplementary Table 3.

For the genes *KCNJ2* and *SCN4AA*, we analyzed the proportion of *com* alleles in individual hybrids. Specifically, we compared the arcsine-transformed proportions between juveniles and adults using a *t* test[35].

## Reporting summary

Further information on research design is available in the Nature Portfolio Reporting Summary linked to this article.

## Data availability

Sequence data have been deposited at NCBI Gene Expression Omnibus under accession GSE240784.

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

## Acknowledgements
The project was funded by the University of Potsdam.

## Author contributions
R.T. conceived and supervised this study and provided financial support. F.C. performed lab work, analysis, and drafted the manuscript with the input from A.B.D., O.B., F.K., and M.D. L.N. performed fish breeding.

## Funding

## Competing interests
The authors declare no competing interests.
