## [Transparent Peer Review file · Communications Biology]

Gene and allele-specific expression during electric organ ontogeny in African weakly electric fish (*Campylomormyrus*)

Corresponding Author: Dr Feng Cheng

Version 0:

Reviewer comments:

Reviewer #1

(Remarks to the Author)

Cheng and colleagues, in manuscript COMMSBIO-24-7312 “Gene and allele-specific expression during electric organ ontogeny in African weakly electric fish”, investigate gene expression differences between two species of weakly electric fish that display differences in electric organ discharge (EOD), between developmental stages and in their F1 hybrids. The authors detect differential expression in 358 genes possibly associated with electric organ development in general (in both species and their hybrids; set C), in 1270 genes possibly related to constancy in EOD during ontogeny (in one species; set B), and 1064 genes possibly related to EOD change during ontogeny (in the other species and their hybrids; set A). The authors then discuss in more detail the hypothetical involvement of a small subset of genes in the 3 gene sets. While the discussion around the selected genes “makes sense”, the authors should clearly address the following:

- 1) it is not very clear why those genes were selected for more in-depth and not others. There is some level of ascertainment bias, as some of these genes have come up in other studies of the same authors. How strong is this bias?
- 2) The authors use fixed SNP differences between parental alleles to examine allele-specific. While this can be justified from a practical point of view, it likely misses important (regulatory and coding) loci responsible for phenotypic differences. Especially considering that this is a young adaptive radiation, it is possible that phenotypic differences are also the result of complex additivity between genes without fixed differences. Have the authors considered the possibility of highly polygenic basis without fixed differences?
- 3) L201-206. If the authors have EOD data from all of these individuals, these data could be entered in a model with gene expression data, rather than simply categorizing juvs, adults, species and hybrids as "short", "medium" and "long"

Minor comments in order of appearance:

- 4) L36-37. Hybridization being a “rare phenomenon in animals” is a rather outdated take, from when many (most?) zoologists still believed that hybridization was an evolutionary dead-end. Please rephrase and update.
- 5) L54. “African weakly electric fishes”
- 6) L98. ~7 cm juvenile: what kind of length is this? Total, standard, fork? Also in L287.
- 7) L143. “For genes where”
- 8) L149-151. For clarity, please write the number of genes
- 9) L175. Maybe “juvenile of ~6–7 cm body length”
- 10) L184. Skeletal
- 11) L194. “in line”
- 12) L261-263. The study of transgressive traits in hybrids is typically done in F2 and in later generation or backcross hybrids, as these will involve the sorting of parental genomes in different proportions. By definition, F1 hybrids will be limited in their scope of phenotypic variation as they have 50% of each subgenome. F1 hybrids are thus more useful for dominance analysis.
- 13) L264. “hybrids proved”
- 14) L267. “alteration. Indeed”
- 15) L313. Was covariance or correlation used for the PCA?

Reviewer #2

(Remarks to the Author)

Cheng et al. perform gene and allele-specific expression analyses in juvenile and adult weakly electric fish, and their

hybrids, to identify genes and their regulatory changes during ontogeny between species that differ in electric organ discharge. The authors identify several potassium-channel related genes that are putatively associated with differences in EOD development between species. Furthermore, their allele-specific expression analyses suggested an overall bias toward the expression of the *C. compressirostris* allele, with only one gene showing a life-stage dependent expression bias toward the *C. rhynchophorus* allele.

Overall, the study is well designed and executed and provides some very interesting insights into the gene regulatory basis underlying differences in electric organ development. The paper is well and clearly written. However, I have some points that would need clarification in my opinion.

General comments:

1) A major concern I have is the lack of control for mapping bias in the allele-specific expression (ASE) analysis. It is well known that the choice of reference genome can drastically impact the mapping rate of specific alleles, which will particularly bias against more divergent alleles (Degner et al. 2009 *Bioinformatics*; Clearly & Seoighe 2021 *Annual Reviews*; Stevenson et al. 2013 *BMC Genomics*; and many more). Since this study used a *C. compressirostris* reference genome, *C. compressirostris* alleles will predominantly map better than *C. rhynchophorus* alleles, potentially leading to the observed allele-specific expression bias toward *C. compressirostris* alleles. While this does not mean that all ASE results are invalid, it might explain the observed bias and change some of the observed result. However, correcting/avoiding mapping bias might also make the observed ASE of *KCNJ2* stronger. I would suggest that the authors repeat the ASE analyses after remapping the data to a reference genome for which all 'divergent' SNPs were masked or use a different approach (e.g. STAR+WASP [DOI: 10.1101/2024.01.21.576391]).

2) To get a better understanding of potential regulatory differences between species and the impact of hybridization on gene expression inheritance in key organs, the authors might be interested in comparing gene expression in hybrids compared to pure-bred species. This might help in identifying genes that show transgressive gene expression patterns (over- and under-dominance) in hybrids, which might point to regulatory differences between parental species. And one might detect species dominance toward one of the parental species. E.g. see Jacobs et al. 2024 *Evolution*; McGirr & Martin 2020 *Molecular Ecology*; Mugal et al. 2020 *Genome Research*). This is really just a suggestion that might provide further interesting insights into gene expression evolution in this really interesting system.

Minor comments:

L.30: I am not sure that this paper really looks at the role of hybridization on complex phenotypic traits, particularly as the authors mention that these species don't tend to hybridize in nature. In my opinion, this paper rather focuses on the regulatory basis underlying the development of complex traits.

L. 36: I don't think I agree that 'successful hybridization is a rare phenomenon' given the growing number of studies on the role of hybridization in speciation and adaptation (e.g. Svardal et al. 2019 *MBE*; Peñalba et al. 2024 *Cold Spring Harbor perspectives in biology*).

L.52: It might be interesting to extend the introduction and the discussion to discuss the findings in the context of context-dependent (e.g. ontogeny) regulatory basis associated with phenotypic divergence. For example, allele-specific expression points toward gene regulatory divergence between species and this might be highly dependent on the context (e.g. developmental stage). This study has the potential to provide interesting insights into how gene regulation associated with a key phenotype throughout ontogeny.

L. 107: Fig2a says 32% but the text here says 35%.

L. 129: 'Up- and down-regulation' are in my opinion probably not the most appropriate terms to use in this context, as one doesn't know as this point if the regulation of the gene has changed. As the authors mention at other points in the manuscript, the size/composition of the electric organ changes throughout development within species, potentially explaining differences in gene expression/abundance. Thus, it might be best to refer to 'expression' or, even more accurate, differences in transcript abundance. However, this is more a suggestion to avoid confusion and not necessary.

L. 179: This observation could potentially be explained by allelic mapping bias and the authors should test for this (see general comment 1).

L. 184 – 186: While the overall PCA shows changes in gene expression relationship that seems to correlate with changes in EOD phenotype, it might be interesting to test if the gene expression inheritance of specific key potassium channels changes in a similar way (e.g. from dominance to parent 1 toward dominance to parent 2 throughout ontogeny), or if they show other patterns of inheritance (e.g. additive expression or overdominance) (see general comment 2).

L.264: "hybridsproved" should be separated.

L.285: Were the same electric organs sampled in juveniles and adults, or were there differences in tissue sampling?

L. 287: Does life-stage correspond to size in both species in the same way? Or are there potential differences that could impact the results? Furthermore, is the sex of these individuals known (in adults and juveniles), and if so, has this been taken into account? Please clarify this, as sex can potentially impact the differential expression analyses.

L.294: What is the minimum RIN value of the samples? This should be provided somewhere.

L.317: "Fold change" rather than "Folder change"

L.324: Which background gene set was used for the GO term analysis? GO term enrichment analysis should only use genes that are expressed in the dataset (after filtering) as the background dataset as otherwise it can drastically skew the results. By default, most analyses use the entire GO dataset (e.g. all genes in the genome) as the background, which might change the over-representation results, as only a subset of genes is usually expressed within a specific tissue. Please describe in detail how the GO term analyses was performed.

L.336: How exactly were allele proportions counted? Was this done with custom scripts or using specific software packages? Please clarify this, as this is crucial for the repeatability of the study.

Version 1:

Reviewer comments:

Reviewer #3

(Remarks to the Author)

Because of the unavailability of the previous reviewer, I have stepped in to examine the response to the previous critiques, specifically by reviewer two. I found that the Reviewer's comments were excellent, and important for improving the study. The author's response responded very nicely to these concerns, and I think greatly improved the paper. I have no additional concerns, and think this exciting paper is worthy of publication.

Reviewers' comments:

Reviewer #1 (Remarks to the Author):

Cheng and colleagues, in manuscript COMMSBIO-24-7312 “Gene and allele-specific expression during electric organ ontogeny in African weakly electric fish”, investigate gene expression differences between two species of weakly electric fish that display differences in electric organ discharge (EOD), between developmental stages and in their F1 hybrids. The authors detect differential expression in 358 genes possibly associated with electric organ development in general (in both species and their hybrids; set C), in 1270 genes possibly related to constancy in EOD during ontogeny (in one species; set B), and 1064 genes possibly related to EOD change during ontogeny (in the other species and their hybrids; set A). The authors then discuss in more detail the hypothetical involvement of a small subset of genes in the 3 gene sets. While the discussion around the selected genes “makes sense”, the authors should clearly address the following:

1) it is not very clear why those genes were selected for more in-depth and not others. There is some level of ascertainment bias, as some of these genes have come up in other studies of the same authors. How strong is this bias?

Response: Thanks for the comments. Of the candidate genes we infer and discuss in detail, actually only the *KCNJ2* had been identified in an earlier study of us, while we infer several new candidates here. To further address this reviewer's comment, we added more analysis, by clustering the gene expression data based on their numerical EOD length (see below). Further, we followed a systematic approach for candidate gene identification which is now outlined in the new chapter in Methods entitled "Identification of candidate genes for EOD duration". The results are presented in the paragraph "Genes related to electric organ development during ontogeny". This inference gave us additional and more credible candidate genes which are all listed in supplementary table 2. The strongest candidates are listed in Table 1, among them electric organ-specific genes as well as EOD duration-specific genes in the adults.

2) The authors use fixed SNP differences between parental alleles to examine allele-specific. While this can be justified from a practical point of view, it likely misses important (regulatory and coding) loci responsible for phenotypic differences. Especially considering that this is a young adaptive radiation, it is possible that phenotypic differences are also the result of complex additivity between genes without fixed differences. Have the authors considered the possibility of highly polygenic basis without fixed differences?

Response: Thank you very much for the comment. There are two relevant issues here, i.e. (1) our focus on fixed SNPs for the ASE inference and (2) the possibility of a potentially highly polygenic basis without fixed differences. Ad (1): Note here that species-differences in gene regulatory elements would yield gene expression differences in downstream target genes among the pure-bred species. These differences are indeed considered in our DEG analysis (results in Figure 2, Table 1 and supplement, regardless of whether these genes carry fixed SNPs. In fact these differences are the major focus of this manuscript. The focus on fixed differences applies only for the ASE analysis. For practical reasons indeed, with our data set we can only discern differential expression among alleles which we can tell apart, i.e., which have some fixed differences among the pure-bred species. This practical constraint does not affect our inference of DEGs (see above), yet for those genes where we can discern among the two parental alleles (and did by our ASE analysis), we receive an additional hint on whether the differences in gene expression may arise from cis- or trans-regulatory elements. Ad (2): EOD divergence is a key trait for speciation, and it exhibits stable differences among species, but almost no variation within species. We are not in a position to disprove a "highly polygenic basis without fixed differences" underlying EOD divergence (as proposed by the reviewer). Identifying such genes would call for GWAS analysis with a reasonable large sample size, beyond the scope of this article. We however humbly argue that the EOD variation patterns (ample variation across species, no variation within species) suggest at least some fixed differences among species. Further, we concentrate here on gene expression differences. The DEG inference is not making any assumption on fixed differences (this constraint applies only for the ASE analysis for practical reasons, see above). Nonetheless, the reviewer is correct in noting that our experimental setup may only allow for identification of underlying major genes, while – in case of a highly polygenic basis of EOD duration modulation – genes with smaller effects may go undetected. To address this reviewer's comment, we now discuss this aspect at the end of the conclusion. The respective statement reads "It should be noted that our DEG/ASE approach is tailored towards identification of candidate genes with strong expression differences and/or fixed differences, relative to divergence in a complex phenotype (here, duration of the electric organ discharge). We argue that – given that the EOD profoundly differs among closely species, but is almost invariant within species of *Campylomormyrus* – the assumption of fixed genetic differences among species, be it in protein-coding exons or gene-regulatory elements, appears reasonable. Indeed, we identify strong candidates for such effects here. Nonetheless, inheritance of the EOD may be polygenic and our approach may overlook involved genes with less profound differential expression and/or without fixed differences among species/EOD phenotypes."

3) L201-206. If the authors have EOD data from all of these individuals, these data could be entered in a model with gene expression data, rather than simply categorizing juveniles, adults, species and hybrids as "short", "medium" and "long".

Response: Thanks for the suggestions. To address this comment, we performed a likelihood ratio analysis to identify the genes whose expression strongly correlated to EOD duration by clustering the gene expression data based on their numerical EOD length. Note that EOD length was almost invariant within cohorts (species and ontogenetic stages), such that – essentially – we yielded only four different length categories for EOD duration, i.e., 0.4ms, 4ms, 5ms, and 40ms

Minor comments in order of appearance:

15) L313. Was covariance or correlation used for the PCA?

Response: We now state that we used covariance for the PCA.

4) L36-37. Hybridization being a “rare phenomenon in animals” is a rather outdated take, from when many (most?) zoologists still believed that hybridization was an evolutionary dead-end. Please rephrase and update.

Response: Thanks for that comment, we omitted the indicated statement of a "rare phenomenon". This part of the introduction was further changed in response to reviewer 2 (see below) to emphasize the potential of our experimental approach to understand the formation of complex phenotypic traits. The respective statement now reads "Context-dependent differential gene expression throughout ontogeny contributes to the formation of complex phenotypes and – hence – divergence in complex traits among closely related species may be understood by investigating differential gene expression during ontogeny. In addition, hybrids among such target species – should they be viable - can play an important role in understanding the emergence of complex phenotypes. Experimental hybrids offer opportunities to study phenotypes that differ from traits in the pure-bred species."

5) L54. “African weakly electric fishes”

Response: Corrected.

6) L98. ~7 cm juvenile: what kind of length is this? Total, standard, fork? Also in L287.

Response: Rephrased to "juvenile of around 7 cm body length"

7) L143. "For genes where"

Response: Corrected

8) L149-151. For clarity, please write the number of genes

Response: Number of genes added.

9) L175. Maybe "juvenile of ~6–7 cm body length"

Response: Corrected.

10) L184. Skeletal

Response: In the course of additional analysis on ASE suggested by Reviewer 2 (see below), this statement became obsolete and was removed.

11) L194. "in line"

Response: Corrected.

12) L261-263. The study of transgressive traits in hybrids is typically done in F2 and in later generation or backcross hybrids, as these will involve the sorting of parental genomes in different proportions. By definition, F1 hybrids will be limited in their scope of phenotypic variation as they have 50% of each subgenome. F1 hybrids are thus more useful for dominance analysis.

Response: Good and valid point, rephrased to "For a particular trait, F1-Hybrids may exhibit intermediate phenotypes, as they express alleles of two subgenomes from their parental species, or may preferentially resemble the trait of one parent (dominance)."

13) L264. "hybrids proved"

Response: Corrected.

14) L267. "alteration. Indeed"

Response: Corrected.

Reviewer #2 (Remarks to the Author):

Cheng et al. perform gene and allele-specific expression analyses in juvenile and adult weakly electric fish, and their hybrids, to identify genes and their regulatory changes during ontogeny between species that differ in electric organ discharge. The authors identify several potassium-channel related genes that are putatively associated with differences in EOD development

between species. Furthermore, their allele-specific expression analyses suggested an overall bias toward the expression of the *C. compressirostris* allele, with only one gene showing a life-stage dependent expression bias toward the *C. rhynchophorus* allele.

Overall, the study is well designed and executed and provides some very interesting insights into the gene regulatory basis underlying differences in electric organ development. The paper is well and clearly written. However, I have some points that would need clarification in my opinion.

General comments:

1) A major concern I have is the lack of control for mapping bias in the allele-specific expression (ASE) analysis. It is well known that the choice of reference genome can drastically impact the mapping rate of specific alleles, which will particularly bias against more divergent alleles (Degner et al. 2009 *Bioinformatics*; Clearly & Seoighe 2021 *Annual Reviews*; Stevenson et al. 2013 *BMC Genomics*; and many more). Since this study used a *C. compressirostris* reference genome, *C. compressirostris* alleles will predominantly map better than *C. rhynchophorus* alleles, potentially leading to the observed allele-specific expression bias toward *C. compressirostris* alleles. While this does not mean that all ASE results are invalid, it might explain the observed bias and change some of the observed result. However, correcting/avoiding mapping bias might also make the observed ASE of *KCNJ2* stronger. I would suggest that the authors repeat the ASE analyses after remapping the data to a reference genome for which all ‘divergent’ SNPs were masked or use a different approach (e.g. STAR+WASP [DOI: 10.1101/2024.01.21.576391]).

Response: Thank you so much for the suggestion! This is an extremely useful comment. Following your suggestion, we re-run the analysis using STAR+WASP. Indeed, the initially inferred preferential expression of *compressirostris* alleles disappeared (and was an artefact of the initially used approach, as suspected by the reviewer). We changed the text accordingly and appreciate that this comment was made before publishing. As the reviewer expected, the relevant ASE pattern of *KCNJ2* remained, such that our major conclusions are not affected.

2) To get a better understanding of potential regulatory differences between species and the impact of hybridization on gene expression inheritance in key organs, the authors might be interested in comparing gene expression in hybrids compared to pure-bred species. This might help in identifying genes that show transgressive gene expression patterns (over- and under-dominance) in hybrids, which might point to regulatory differences between parental species.

And one might detect species dominance toward one of the parental species. E.g. see Jacobs et al. 2024 Evolution; McGirr & Martin 2020 Molecular Ecology; Mugal et al. 2020 Genome Research). This is really just a suggestion that might provide further interesting insights into gene expression evolution in this really interesting system.

Response: Thank you for this thoughtful comment. We addressed this comment by comparing gene expression patterns between (1) each of the parental pure bred species and (2) the hybrids (significant differences reported in supplementary table 3). We did not observe any indication of overdominance or underdominance in gene expression among our candidate genes. For *KCNJ2*, the gene with strong ASE towards the rhy allele, gene expression in the hybrid was higher than in *C. compressirostris*, but lower than in *C. rhynchophorus*, further corroborating our inference of cis regulatory differences among the com and the rhy allele at this gene.

Minor comments:

L.30: I am not sure that this paper really looks at the role of hybridization on complex phenotypic traits, particularly as the authors mention that these species don't tend to hybridize in nature. In my opinion, this paper rather focuses on the regulatory basis underlying the development of complex traits.

Response: Thanks for the comment. We have rephrased the introduction part. Instead of hybridization, we are more emphasize the role of experimental hybridization on the study of phenotypic evolution.

L. 36: I don't think I agree that 'successful hybridization is a rare phenomenon' given the growing number of studies on the role of hybridization in speciation and adaptation (e.g. Svardal et al. 2019 MBE; Peñalba et al. 2024 Cold Spring Harbor perspectives in biology).

Response: Thanks for the comment, we have rephrased it.

L.52: It might be interesting to extend the introduction and the discussion to discuss the findings in the context of context-dependent (e.g. ontogeny) regulatory basis associated with phenotypic divergence. For example, allele-specific expression points toward gene regulatory divergence between species and this might be highly dependent on the context (e.g. developmental stage). This study has the potential to provide interesting insights into how gene regulation associated with a key phenotype throughout ontogeny.

Response: Thanks for the comments, we added these aspects to our introduction and discussion on the phenotypic ontogeny.

L. 184 – 186: While the overall PCA shows changes in gene expression relationship that seems to correlate with changes in EOD phenotype, it might be interesting to test if the gene expression inheritance of specific key potassium channels changes in a similar way (e.g. from dominance to parent 1 toward dominance to parent 2 throughout ontogeny), or if they show other patterns of inheritance (e.g. additive expression or overdominance) (see general comment 2).

Response: Thank you for this comment. In fact, this information is already provided in supplementary table 3 which lists the genes with fixed allelic differences and an expression proportion change between juveniles and adults over 0.1. Among those genes, we only found one potassium channel gene (KCNJ2). However, we identified the sodium channel gene SCN4aa, which is – with regard to EOD production - considered the most important sodium channel gene among different electric fishes. The alleles from both parental species were almost equally expressed in juveniles (com proportion = 0.55), while adults showed expression dominance towards the allele of the short EOD parental species *C. compressirostris* (com proportion = 0.67).

L.285: Were the same electric organs sampled in juveniles and adults, or were there differences in tissue sampling?

Response: The method of tissue sampling was the same in juveniles and adults. In both ontogenetic stages, the electric organ is located in the caudal panel. The only difference is the smaller size of the electric organ in juveniles, relative to adults.

L. 287: Does life-stage correspond to size in both species in the same way? Or are there potential differences that could impact the results? Furthermore, is the sex of these individuals known (in adults and juveniles), and if so, has this been taken into account? Please clarify this, as sex can potentially impact the differential expression analyses.

Response: Length in these fish depends both on age and rearing conditions. The adult *C. compressirostris* and *C. rhynchophorus* as well as their hybrids were of similar size and had reached the stage of sexual maturity. The juveniles were of equal size (~7cm). The time for them to develop to a ~7cm head-body length took approximately 6 month from fertilized embryos. We cannot guarantee that the juveniles were exactly in the same ontogenetic stage, however, relevant to this study, we had our relevant phenotypic trait (the EOD) measured in these juveniles, such that differences in ontogenetic stage (if any) may have introduced some

variation into gene expression, but should not systematically bias our inference of EOD-related genes.

The sex is difficult to know before the fish reach maturity. In the adult samples (originating from Cheng et al 2024 MBE), both male and female were included in the analysis. Again, gene expression likely varies to some degree among the sexes, yet our focus trait, the EOD, does not exhibit differences between males and females in *Campylomormyrus*, hence our inference of EOD-related candidate genes should not be affected by inclusion of specimens from both sexes.

L.294: What is the minimum RIN value of the samples? This should be provided somewhere.

Response: Thanks for the suggestion. The samples with RIN over 9 were used in the analysis, we clarified this aspect in the manuscript.

L.324: Which background gene set was used for the GO term analysis? GO term enrichment analysis should only use genes that are expressed in the dataset (after filtering) as the background dataset as otherwise it can drastically skew the results. By default, most analyses use the entire GO dataset (e.g. all genes in the genome) as the background, which might change the over-representation results, as only a subset of genes is usually expressed within a specific tissue. Please describe in detail how the GO term analyses was performed.

Response: Thanks for the comment. We performed this analysis with the NCBI tool DAVID where we chose the zebrafish background, the most closely related species available. Admittedly, we have issues in fully understanding this reviewer's comment. It appears to us that – if we would also use the genes of our data set as background - we would in fact be unable to detect any enrichment, as the enrichment analysis indeed looks for enrichment in a data set, relative to an independent background. We humbly argue that we have applied an established approach in an appropriate manner, but please provide clarification, if you still have concerns regarding our inference.

L.336: How exactly were allele proportions counted? Was this done with custom scripts or using specific software packages? Please clarify this, as this is crucial for the repeatability of the study.

Response: Thanks for the comment, we added more details in the Method part. The allelic depth was calculated in BCFtools and we now provide the commands used. Filtering for low quality and low allelic depth was performed by simply sorting in excel.

L. 107: Fig2a says 32% but the text here says 35%.

Response: Corrected to 32%.

L. 129: ‘Up- and down-regulation’ are in my opinion probably not the most appropriate terms to use in this context, as one doesn’t know at this point if the regulation of the gene has changed. As the authors mention at other points in the manuscript, the size/composition of the electric organ changes throughout development within species, potentially explaining differences in gene expression/abundance. Thus, it might be best to refer to ‘expression’ or, even more accurate, differences in transcript abundance. However, this is more a suggestion to avoid confusion and not necessary.

Response: Thanks. The indicated statement has been rephrased in the course of additional analysis suggested by reviewer 1 (cluster analysis on EOD length). Generally, although we conceptually agree to the reviewer, speaking about differences transcript abundances (instead of up-/down-regulation) might rather irritate than inform readers. We carefully re-read the manuscript to ensure that the usage of our terms is consistent and can be correctly understood.

L. 179: This observation could potentially be explained by allelic mapping bias and the authors should test for this (see general comment 1).

Response: Yes, you were correct (see above) and – after performing the suggested analysis – the general bias between species disappeared, hence this statement was omitted.

L.264: “hybridsproved” should be separated.

Response: Corrected.

L.317: “Fold change” rather than “Folder change”

Response: Corrected.